# Agent-based model projections for reducing HIV infection among MSM: Prevention and care pathways to end the HIV epidemic in Chicago, Illinois

Wouter Vermeer[1,2,3]*, Can Gurkan[1,3,4], Arthur Hjorth[5], Nanette Benbow[1], Brian M. Mustanski[1,6,7], David Kern[8], C. Hendricks Brown[1], Uri Wilensky[2,3,4,9]

**1** Center for Prevention Implementation Methodology for Drug Abuse and HIV (Ce-PIM), Feinberg School of Medicine, Northwestern University, Chicago, IL, United States of America, **2** Northwestern Institute for Complex Systems (NICO), Northwestern University, Evanston, IL, United States of America, **3** Center for Connected Learning and Computer-Based Modeling (CCL), Northwestern University, Evanston, IL, United States of America, **4** Department of Computer Science, McCormick School of Engineering, Northwestern University, Evanston, IL, United States of America, **5** Department of Management, Aarhus University, Aarhus, Denmark, **6** Institute for Sexual and Gender Minority Health and Wellbeing, Northwestern University, Chicago, IL, United States of America, **7** Department of Medical Social Sciences, Northwestern University Feinberg School of Medicine, Chicago, IL, United States of America, **8** Chicago Department of Public Health (CDPH), Chicago, IL, United States of America, **9** Learning Sciences Program, School of Education and Social Policy, Northwestern University, Evanston, IL, United States of America

\* Wouter.vermeer@Northwestern.edu

**Editor:** Hamid Sharifi, HIV/STI Surveillance Research Center and WHO Collaborating Center for HIV Surveillance, Institute for Future Studies in Health, Kerman University of Medical Sciences, IRAN, ISLAMIC REPUBLIC OF

## Abstract

Our objective is to improve local decision-making for strategies to end the HIV epidemic using the newly developed Levers of HIV agent-based model (ABM). Agent-based models use computer simulations that incorporate heterogeneity in individual behaviors and interactions, allow emergence of systemic behaviors, and extrapolate into the future. The Levers of HIV model (LHM) uses Chicago neighborhood demographics, data on sex-risk behaviors and sexual networks, and data on the prevention and care cascades, to model local dynamics. It models the impact of changes in local preexposure prophylaxis (PrEP) and antiretroviral treatment (ART) (ie, levers) for meeting Illinois' goal of "Getting to Zero" (GTZ) — reducing by 90% new HIV infections among men who have sex with men (MSM) by 2030. We simulate a 15-year period (2016-2030) for 2304 distinct scenarios based on 6 levers related to HIV treatment and prevention: (1) linkage to PrEP for those testing negative, (2) linkage to ART for those living with HIV, (3) adherence to PrEP, (4) viral suppression by means of ART, (5) PrEP retention, and (6) ART retention. Using tree-based methods, we identify the best scenarios at achieving a 90% HIV infection reduction by 2030. The optimal scenario consisted of the highest levels of ART retention and PrEP adherence, next to highest levels of PrEP retention, and moderate levels of PrEP linkage, achieved 90% reduction by 2030 in 58% of simulations. We used Bayesian posterior predictive distributions based on our simulated results to determine the likelihood of attaining 90% HIV infection reduction using the most recent Chicago Department of Public Health surveillance data and found that projections of the current rate of decline (2016-2019) would not achieve the 90% (p = 0.0006) reduction target for 2030. Our results suggest that increases are needed at all steps

**Data Availability Statement:** Data are model code are available from: https://www.comses.net/ codebases/05ad0cc2-d63e-4e4f-beba-4ea2e85a5e81/releases/1.0.3/ And contains separate files for the following: - Full model code of the simulation model used in this paper - Code used for the large scale experiments conducted in this paper - Datasets containing the results for each of the large scale experiments conducted in this paper - Analysis scripts converting the result data into the the figures/tables presented in this paper. This should allow any who is willing, to reproduce any/all of the steps of the analysis process presented in the paper.

**Funding:** This work has in part been sponsored by the following grants: HB - National Institute on Drug Abuse funded Center for Prevention Implementation Methodology (P30DA027828) BM - National Institute on Drug Abuse funded grant on Multilevel Influences on HIV and Substance use in a YMSM Cohort (U01 DA036939) DK - Health Resources and Services Administration Ryan White HIV/AIDS Program Part A Emergency Relief for Areas with Substantial Need for Services (CFDA #: 93.914) DK - Centers for Disease Control and Prevention Integrated HIV Surveillance and Prevention Programs for Health Departments (CFDA #: 93.940) DK - Department of Housing and Urban Development Housing Opportunities for Persons with AIDS (CFDA #: 14.241) UW - National Science Foundation funded grant on Adding Computational Thinking Components to the High-School Science Curriculum to Broaden Participation in Computational Science (NSF-STEM +C-1842374) The funders had no role in study design, data collection and analysis, decision to publish, or preparation of the manuscript.

**Competing interests:** The authors have declared that no competing interests exist.

of the PrEP cascade, combined with increases in retention in HIV care, to approach 90% reduction in new HIV diagnoses by 2030. These findings show how simulation modeling with local data can guide policy makers to identify and invest in efficient care models to achieve long-term local goals of ending the HIV epidemic.

## Introduction

Over the past decade, advances in the development of evidence-based HIV prevention and care interventions have led to considerable declines in new HIV infections in the United States. These declines, coupled with a bold national HIV/AIDS strategic plan [1, 2] and the recent Ending the HIV Epidemic (EHE) initiative [3], have defined a vision of HIV elimination throughout the United States. Using pre-exposure prophylaxis (PrEP) and antiretroviral treatment(ART), the national EHE initiative seeks to reduce new HIV infection diagnoses in the United States by 75% in 2025 and by 90% by 2030.

While EHE plans throughout the country share similar goals, the starting point and pathway to achieving these goals vary considerably from one jurisdiction to another. To date, 13 states and 23 local jurisdictions have developed EHE plans; many more are in the process of development [4]. According to the most recent progress report from the Centers for Disease Control and Prevention [5], intermediary 2020 targets of decreasing new HIV infections by 25% were only met by 7 states, with 13 states moving in the direction of the goal. A separate goal of increasing viral suppression among those diagnosed with a HIV infection has not been met by any state, although 26 are making progress toward it. For PrEP, the initial goal of a 500% increase in prescriptions has been far exceeded [6]; however, in 2019 only 23.4% of those eligible were prescribed PrEP. In addition, sex, race and age disparities in PrEP coverage remain [5]. Similar disparities exist in retention in HIV care and viral suppression among HIV-positive individuals [5], highlighting the necessity to differentiate intensities and combinations of interventions in each jurisdiction in order to reach the EHE goals for all.

Recently, a number of HIV modeling studies have explored the impact of various interventions on reducing HIV infections. Shah et al [7] developed a dynamic transmission model of HIV progression and care engagement to determine which steps of the HIV care continuum had the greatest impact on viral suppression rates and reductions in new HIV infections nationally, focusing on awareness of HIV serostatus and linkage to and retention in HIV care. They found that increases in HIV testing or linkage to care in isolation had a moderate impact on disease burden, whereas increases in HIV care retention led to a 20% reduction in new HIV infections. An agent-based model (ABM) developed by Jenness and colleagues [8] for men who have sex with men (MSM) in Atlanta showed that neither a 10-fold increase in screening nor a 10-fold increase in retention in HIV care would achieve local EHE goals, whereas a combination of these interventions would achieve a 90% reduction in 12 years. Another study by Nosyk and colleagues [9] used economic modeling to examine a combination of evidence-based prevention and care interventions in 6 US cities to identify the highest-value intervention combination to reduce HIV infections. They found that each city had a unique health-maximizing combination of interventions to reach EHE goals and achieved varying levels of reduction in HIV infections (range, 39.5%-60.7%). These studies suggest (1) that jurisdictions are faced with identifying unique combination and levels of interventions needed to maximize reduction of new HIV infections locally and (2) that locally grounded

simulation models can help by identifying intervention impact, and as such provide quantitative guidance in the goal setting and execution of local EHE plans.

In this article, we describe how we support the Chicago Department of Public Health (CDPH) implementation of Illinois' EHE plan, known as the "Getting to Zero Illinois (GTZ)" [10]. GTZ, in line with EHE goals, aims to reach a 90% reduction in new HIV infections by 2030. While the aim is clearly defined, the path to achieving it —the proposed interventions, the optimal combinations and levels of PrEP and ART intensity—has yet to be delineated. Traditional clinical cannot provide intervention evidence because of their limited time span, emphasis on internal validity over external contextual forces, and incomplete representation of populations impacted by HIV [11–13]. This has resulted in a call for systems science methods to better understand and predict systemic outcomes in health care [14–19] and in a specialty field such as HIV prevention [20]. More specifically, there is a need to account for individual variables, interactions that occur between individuals, feedback loops, and uncertainty about behavior at the system level, all of which results in heterogeneity across healthcare settings and individuals. Such heterogeneity drives local dynamics and is required to capture phenomena like health disparities. To address this, our modeling relies on ABMs—a computer simulation methodology that embraces heterogeneity in behaviors and interaction of individual, allows for emergence of systemic behaviors, and can extrapolate into the future [19, 21]—to capture the full complexity of the social system dynamics.

ABM have traditionally been used to build theory and make conceptual claims about the dynamics of complex systems [22]. However, more recently, ABMs are being adopted as a tool to support decision making in social systems. To help support decision making in the context of HIV prevention, ABMs must capture local context and dynamics [19] and incorporate details that affect a phenomenon in the real world [23]. Capturing realistic behaviors, comes with a strong dependance on local data, and a need for validation. Population-level dynamics should be aligned with those observed in reality (e.g., short-term incidence rates) and individual- (agent-) level behaviors should be aligned with local data [22].

We present an ABM—the Levers-of-HIV-Model (LHM) [24], that simulates 15 years of new HIV infections among MSM. The model explores the impact of various scenarios of change in the PrEP and ART cascades, which we refer to as *levers*, as a tool for examining trends in new HIV infections in Chicago. These 6 levers focus on delivery of evidence-based prevention and treatment interventions: (1) linkage to PrEP for those testing negative, (2) linkage to ART for those living with HIV, (3) adherence to PrEP, (4) viral suppression by means of ART, (5) PrEP retention, and (6) ART retention. Our simulations can be used to explore, tailor, and optimize implementation strategies that target these levers of change and as a support tool to inform policy makers by providing actionable, cost-efficient, and timely information on which to implement strategies that meet GTZ goals. We use our model to identify optimal combinations that would prevent committing scarce resources into implementing less impactful strategies.

We review existing ABMs for HIV; describe our model and how local data have been used to inform it; describe model validation and the structure of the simulation experiment; present experiment outcomes and how they relate to GTZ 2025 and 2030 goals; and describe how model results can inform public health implementation strategies.

## ABMs for HIV spread

The epidemic nature of HIV spread, the impact of heterogeneous sex-risk behaviors, and the complex social systems in which spread occurs make ABMs ideal tools for predicting optimal treatment and care strategies to end the HIV epidemic. Various groups have developed high-

**Table 1. Comparison of agent-based models for HIV transmission.**

| Model | Scope | Population | Interaction type | Network data | Are model outcomes fitted | Interventions considered |
|-------|-------|-----------|------------------|--------------|---------------------------|--------------------------|
| Titan | NY metro | All | IDU/Sexual | Estimated | Yes | None |
| Path 2.0 | US | PLWH | Sexual | Aggregate | No | None |
| Epimodel | Atlanta | MSM | Sexual | Local, Aggregate | Yes | Radiation: ART, Screening Transmission: Condom use Reception: PrEP |
| Bars2.0 | Chicago south-side | YB-MSM | Sexual | Local, Aggregate | Yes | Radiation: ART Transmission: N/A Reception: PrEP |
| LHM | Chicago | MSM | Sexual | Local, Individual | No | Radiation: ART Transmission: N/A Reception: PrEP |

**Notes:** ART indicates antiretroviral treatment; IDU, injection drug user; MSM, men who have sex with men; YB, Young Black; NA, not applicable; PLWH, people living with HIV; PrEP, preexposure prophylaxis.

fidelity models to support decision making aimed to achieve this goal. The most prominent and comprehensive models are the Marshall model [25], the PATH2.0 model [26], EpiModel [27], and the BARS model [28]. We compared these models with the Levers of HIV model [24] (Table 1). While each model is grounded in field data and aims to capture HIV spread dynamics, they differ in terms of the social and sexual contexts they incorporate and the means of implementation.

The Marshall model [25, 29], later labeled TITAN (Treatment of Infection and Transmission in Agent-Based Networks), was one of the first HIV-focused high-fidelity ABMs. Written in Python, it specifically addresses the cross-section of injecting and non-injecting drug use and HIV spread in the New York metropolitan area. Individual-level heterogeneity is present in this model; however, local data on sex-risk behavior and drug-use interactions is scarce, and this model assumes homogeneous mixing among its populations and relies on parameter fitting to ensure matching between model and reality of observable emergent characteristics.

The PATH 2.0 project similarly incorporates individual-level properties but bases its sexual interaction dynamics on a sample of national US data for people living with HIV. As such, it models national-level dynamics. Similar to the TITAN model, PATH 2.0 uses parameter fitting to align model results to reality; however, this model incorporates aggregate-level interaction data to build heterogeneous interaction networks. The national data and model scope of PATH 2.0 likely miss critical local dynamics and sexual networks, the latter of which have been shown to affect the dynamics of spreading phenomena [30].

Epimodel 2.0 [8, 27, 31], written in R, provides a simulation platform for diverse infectious diseases. For its modeling of HIV dynamics, Epimodel uses social and sexual dynamics from 2 cohort studies among MSM in Atlanta, Georgia. It focused on local dynamics and considering the disproportionately affected MSM population. Heterogeneity in the sexual networks is explicitly modeled based on aggregate summaries of network data (degree distributions and network homophily). Epimodel uses limited parameter fitting to align the modeled dynamics to ecological-level dynamics, using a single parameter that represents an inflation of the number of sex acts within the network ties formed. Various extensions of Epimodel exist, one of which has been tailored to fit Seattle ecological dynamics [32].

The BARS and related ABMs [28], written in C++, R (statnet [33]) and Repast, use data from young Black MSM in Chicago and combine a representative cohort with specific data of those who have had criminal justice involvement, a group that is disproportionately affected

by the HIV epidemic [34]. It uses detailed local egocentric interaction network data to model the formation and ending of sexual relationships and relies on parameter fitting to align model results to observed systemic trends.

The LHM model presented in this article [24] is written in NetLogo [35] and differs from previously presented models. First, this work is derived from a partnership between the local health department responsible for administering HIV programs—the Chicago Department of Public Health (CDPH)—and researchers at Northwestern University and focuses on the model on Chicago. Second, it uses longitudinal sexual network data from a large, ongoing observational study in Chicago [36–38] to model formation and dissolution of sexual pairings, rather than basing such dynamics on system-level aggregates. Third, this model limits its use of parameter fitting to inputs rather than outputs, ensuring that system-level dynamics emerge from the bottom up and the model does not rely on parameter fitting to align modeled system-level dynamics with observed real-world trends. Last, the model combines both the treatment (ART) and prevention (PrEP) intervention levers into a single model, allowing these levers to be considered simultaneously and interaction effects across and within them to be explored.

Most HIV-related ABMs have been used to conduct virtual experiments of the impact of various interventions [29, 31, 39, 40]. These interventions, like all interventions in propagating phenomena, can be classified into targeting 1 of the 3 sub-processes of propagation [41]: radiation, transmission, and reception. In the context of HIV spread, reducing radiation targets infectiousness through detecting and reducing population viral load. Reducing transmission includes policies such as safe sex campaigns. Reducing reception addresses susceptibility, which includes providing PrEP to eligible individuals. For each intervention target, the potential impact is conditional on sexual networks, local spreading dynamics, and baseline levels for any of these targets, all of which are highly heterogeneous across settings. This makes the value of ABMs for supporting decision making dependent on their ability to capture local dynamics [19, 28].

## Methods

Herein we describe the model Levers of HIV model itself and our analysis plan, including model validation and experimental design.

### The Levers of HIV Model (LHM)

Currently, the LHM focuses on Chicago's most heavily impacted population: MSM, who account for most (73.5%) of new HIV diagnoses in Chicago. Unprotected anal intercourse accounts for 95% of transmission and nearly all [42] (96%) new cases within this group. By focusing on this group and this mode of transmission, we significantly simplify the model while prioritizing the dominant population and means of spread for HIV.

The LHM consists of 5 major modules: (1) demographics, (2) network dynamics and partner selection, (3) HIV transmission and progression, (4) health disparity, and (5) prevention and care. An overview and description of the role of local data is provided for each module. A comprehensive and detailed description of the models behaviors is provided in the Supplementary Information (S1 Appendix), along with the model code [24].

**Demographics module.** *Motivation for inclusion.* To accurately describe the characteristics and dynamics of our specific context we need to ensure that the population in our model is representative of this context. For our model, we want racial, spatial, age, and HIV status distributions to follow those observed among MSM in Chicago.

*Means of inclusion and data sources.* Our starting point is the numbers of males aged 13 to 70 years in the Chicago census data by race/ethnicity. Based on the AIDSVu [43] MSM

estimates of 6.6% of males in Cook County aged 13 to 80 years old, we calculated a total of approximately 65,000 MSM living in Chicago. As detailed local data is missing, our model assumes that the proportion of MSM is constant across race/ethnicity and age. To reduce computational requirements, the LHM samples 10% of this population (6500 individuals), describing the city on a 1:10 scale while maintaining the race/ethnicity, age, and location data from the Chicago census, and HIV prevalence distributions using demographic conditions from CDPH surveillance data.

The population size in the model will fluctuate over time. Model dynamics account for individuals who die (both due to natural causes and to HIV) at rates based on local data for non-HIV and HIV-related mortality and for individuals aging out of our model's age range or entering our model as they become of age. Our model does not account for in or out migration as this factor was not notable; inflow and outflow were roughly equal in our model, keeping the total population around 6500 throughout the simulation.

**Network dynamics and partner selection module.**    *Motivation for inclusion.* The average rates of interactions, the variance in those rates, and the local structures that form due to this variance impact potential spread. To capture such heterogeneity, data on sexual networks and interactions from the RADAR longitudinal cohort study of more than 1200 Chicago MSM assessed over a period of 3 years at 6-month intervals [37] was used to capture individual level partnering behaviors, which inform individual-level partnering preference and sexual behaviors in the model. We let the network structure emerge from this process, rather than generating individual behaviors from global network attributes, as done in other models (Titan, PATH 2.0, Epimodel, and Bars2.0). The RADAR cohort is primarily focused on younger MSM (ages 16–34 years), and information on older MSM's partnering patterns is scarce. We use age assortative mating trends in the RADAR data to extrapolate the behavior of the older MSM in our model. While the sexual networks of the comparatively older MSM are not fully represented in our input data, MSM younger then 40 years account for roughly 78% of new HIV infections among MSM in Chicago.

*Means of inclusion and data sources.* Agents (i.e. Individuals) in the LHM form and end sexual ties, resulting in sexual networks that are dynamic over time. Our model distinguishes between one-time sexual events, ties that last less than a single week, and ties that persist for a longer period, each with different attributes (e.g., condom use). Ties (and resulting networks) are formed based on agent-level preferences for sexual activity and attributes of the desired partner. This process is based on optimizing dyadic fit based on both partners' HIV-status, race/ethnicity, age, and sex-role (see S1 Appendix for details). Once ties are formed, we generate longitudinal profiles of tie duration, rate of sex acts, and rate of condom use assembled from the observed distribution from the RADAR study. RADAR data provide rates of partner formation, correlations between ego and alter attributes, and the distributions of sexual behaviors and characteristics within ties once they are formed.

**HIV transmission and progression module.**    *Motivation for inclusion.* This module represents the probability of transmitting HIV in sero-discordant pairs during a given sexual act. Transmission can be subdivided into 3 sub-processes of radiation, transmission, and reception [41]. The transmission of HIV depends on the seropositive individual's viral load (radiation), the seronegative individual's genetic receptibility and use of PrEP (reception), as well as characteristics of the sex act including sexual position and protective vs. unprotective anal intercourse (transmission). To realistically capture the risk of transmission during sex, we capture each of these elements in our model.

*Means of inclusion and data sources.* The viral load of seropositive individuals is known to have different trajectories in the absence or presence of treatment. Based on existing literature on the virality of HIV [44, 45] the viral load is modeled as a function of time and ART. Our

model follows the viral load progression module introduced in Epimodel [31], which, in the absence of ART, is described by a 4-stage process of acute rise, acute fall, stable setpoint levels, and rise during the AIDS stage until death occurs (see S1 Appendix for details). The probability of radiation is determined as a function of the viral load during the sex act. For determining the likelihood of transmission, we use RADAR local data to determine the rate of and variation in condom use and sexual positions, and consider the circumcision of the seronegative partner to determine the chances of viral infection. We model the presence of CCR mutations (which account for large reductions in susceptibility—70% or 100%, depending on mutation type) based on national rates by race/ethnicity [46] and PrEP use based on RADAR data, resulting in likelihood of reception. We then take the product of radiation, transmission, and reception to determine a total risk of transmission for a given sex act.

**Health disparity module.**   *Motivation for inclusion.* African Americans and Latinos, as well as young MSM, experience huge disparities in HIV infection risk. Evidence from Chicago and other cities shows that race and ethnic disparities are not due to higher-level risk behaviors (e.g., unprotected anal intercourse) [37] but are functions of differential social determinants, including higher exposure to community viral load and limited health care access. Racial segregation is particularly prevalent in Chicago, and socioeconomic characteristics, access to healthcare, and community-level stigma vary significantly from neighborhood to neighborhood. These sociodemographic and community-level factors, which have been shown to be associated with poor HIV-related outcomes and racial/ethnic disparities [47], stem from the environment in which individuals live rather than their individual behaviors or characteristics. While such disparities are known to be present, the exact mechanisms by which they occur are not fully understood.

*Means of inclusion and data sources.* Chicago reports community-level health, care, and prevention data by its 77 community areas, which have been studied by researchers consistently since the 1920s [48, 49] and used by CDPH to survey and allocate resources. To capture some of these disparities in our model, we include a function that is based on included community area viral load (defined as the community HIV prevalence multiplied by community rate of nonsuppression) and the neighborhood hardship index as a proxy for community risks. This proxy captures major effects of the regional health disparities observed, while leaving the question of what drives these disparities for future research. We use this proxy of community risk by adjusting each agent's risk depending on his neighborhood of residence. The risk of contracting HIV during sex in a serodiscordant tie act will increase or decrease by a given factor based on the local viral load and overall hardship in that neighborhood.

**Prevention and care module.**   *Motivation for inclusion.* Treatment and prevention are 2 primary ways in which the care system attempts to impact the spread of HIV. First, treatment can reduce transmission (radiation) by having seropositive individuals receive ART. This approach, referred to as Treatment as Prevention (TasP), aims to reduce the viral load of these individuals, directly translating into lowering the likelihood for these individuals to transmit HIV, thereby preventing new infections. Second, prevention can impact the chances of contracting HIV (reception) for seronegative individuals. By having them receive PrEP, a medication that reduces the risks of seroconversion for seronegative individuals who come into contact with HIV, the chances of spread are reduced.

*Means of inclusion.* The prevention and care module in LHM, summarized in Fig 1, includes a prevention arm, involving PrEP for seronegative individuals (prevention, green), and a treatment arm, involving TasP for people living with HIV (red). For each arm, individuals will cycle through being linked to the appropriate medication, being adherent (for PrEP) or suppressed (for TasP), and being retained in care, resulting in individuals over time either being partly or fully protected (if negative) or being virally suppressed (if positive).

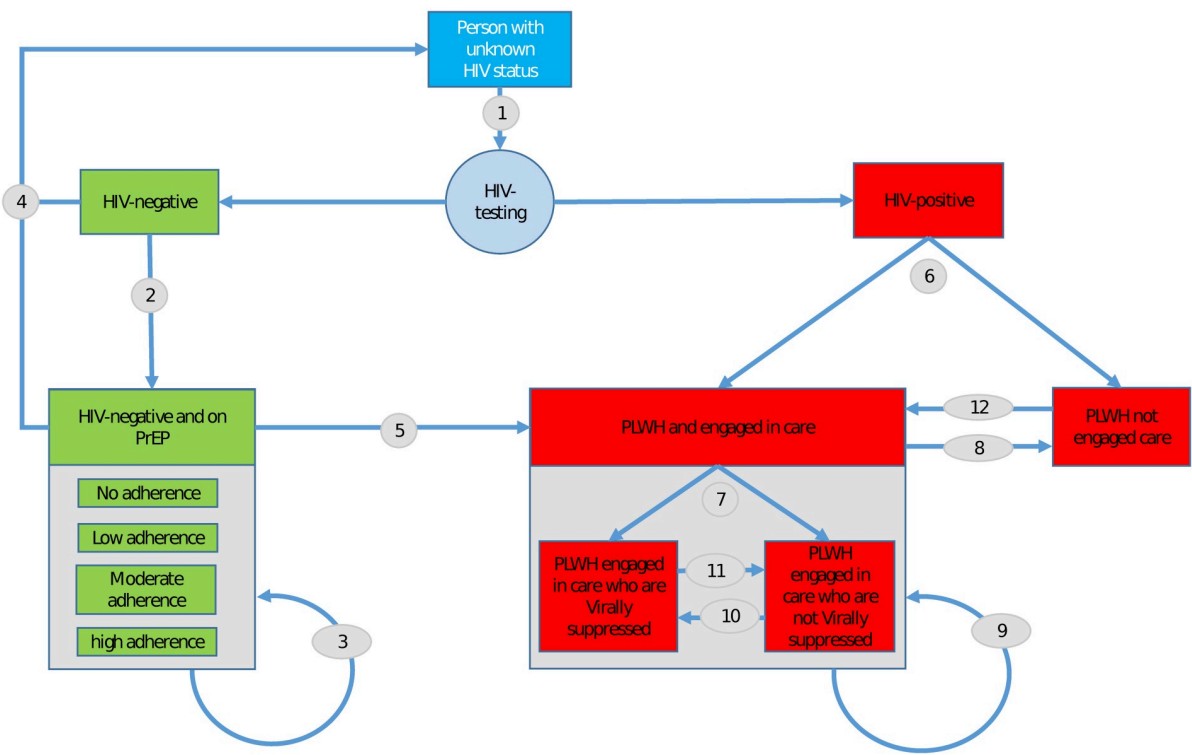

**Fig 1. The care and treatment system.** A flowchart of the stages of the HIV treatment cascade that agents in the LHM can go through.

Each of the flows in Fig 1 represents a rate or probability that is based on field data. All rates in this figure are based on local data, with the exception of the rate of testing. As local data on rates of testing was unavailable for Chicago, we opted to align our modeled testing rate to previously published Epimodel [31]). To inform the remainder of TasP-related rates surveillance data provided by the CDPH was used, while for PrEP rates, RADAR cohort data was used. The Supplementary Information provides details on these rates (S1 Appendix).

**Parameter fitting.** Most parameters included in the LHM are known from epidemiological modeling (e.g., infectivity per sex act) or from local data (e.g., distribution of sex partners). We call these *fixed* parameters. In contrast, parameters in the model that are adjusted to fit aspects of the model are called *free* parameters. In the LHM we use free parameter fitting for 2 specific sub-modules where data are limited. First, we used free parameters to fit HIV incidence rates by race and ethnicity in the health disparity module. We found an appropriate weighting to ensure that 2 community-level risk factors of viral load and hardship agreed with the 1-year race and ethnicity incidence rates. Second, we used free parameters to fit the treatment module. Our initial TasP modeling was based on data that only covered the first 2 years of viral suppression after diagnosis. When discussing with our partners at CDPH, this was seen as inadequate for those with longer histories of HIV positivity. To remedy this and to account for long-term viral suppression, we adopted a rate capturing a tendency to (temporarily) discontinue care for those diagnosed more than 2 years earlier. Because no direct data were available to inform this rate, we fit this parameter based on minimizing the error in reproducing the known levels of suppression and levels of active treatment among people living with HIV. While there is uncertainty on the accuracy of this rate, we performed a sensitivity analysis with alternative rates to examine whether our fitting assumption impacted our results.

## Analysis plan

The analysis of model behavior involves 2 steps. The first checks whether the model is properly validated, and whether systemic behavior matches what is observed in the real world. The second uses this validated model to run a virtual experiment by projecting new infection among MSM in Chicago to 2030 using varying levels of interventions in the HIV prevention and care cascades.

## Model validation

The dynamics of each of the modules within LHM are based on field data, ensuring that agent-level behavior agrees with observed trends. While the use of local data in each module independently should yield realistic dynamics, we ensured that interaction and accumulation of modules yielded systemic behaviors that are in line with observed trends in field data.

A primary aim is to discover optimal pathways to reduce new infections through a combinations of levers; therefore annual HIV incidence is a key system level outcome to validate against. Because our model is built using 2015 data, we test the extent to which the model aligns with the HIV incidence observed in 2016, for which we have CDPH surveillance data. We checked whether the model simulations produced MSM incidence rates consistent with real-world data and also compared incidence distributed by age and incidence distributed by race/ethnicity.

## Experimental design

As we intend to support the CDPH's decision making to achieve GTZ goals in Chicago, we focus our experimental design on levers of change that indirectly follow from the jurisdiction's health system funding of programs to recipient organizations and its own activities. Based on the care system modeled in LHM, we identify 6 levers of change that pertain to proximal factors in delivering evidence-based interventions: (1) linkage to PrEP for those testing negative, (2) linkage to ART for those living with HIV, (3) adherence to PrEP, (4) viral suppression by means of ART, (5) PrEP retention, and (6) ART retention. Levers are not implementation strategies themselves but targets for monitoring selected strategies. For each lever we considered various levels using a range from the "current" 2015 levels as a baseline (identified as level 0) to what is considered maximally achievable (Table 2).

**Linkage to PrEP care.** Linkage captures the extent to which individuals with a negative test result initiate PrEP. Based on RADAR's rate at which individuals adopt PrEP, we calculate the baseline probability of PrEP uptake for the LHM. The linkage to PrEP lever then annually increases this rate. Using 5 levels, this rate increased by either 0, 2, 4, 6, 8, 10 absolute percent annually, which in the most extreme scenario would yield 100% linkage after 10 years. Uptake encompasses all the steps that precede initiation (e.g., gaining awareness of and access to PrEP, as well as a negative HIV test); hence, achieving these increases in uptake would require multiple strategies.

**Adherence to PrEP.** The extent to which PrEP will be effective in reducing risk of HIV acquisition is conditional on the individual's adherence (see S1 Appendix). At the baseline level (level 0), individuals are distributed among 4 adherence groups, based on observed rates. This distribution is then increased for each of the other levels. At level 1 it is assumed no individuals will be completely non-adherent, and the individuals that were previously non-adherent will be evenly divided among the remaining adherence groups. At level 2 the assumption is everyone will be fully adherent.

**Retention in PrEP care.** Retention captures the tendency to stay in PrEP care over time. PrEP prescriptions are provided for a limited duration to allow for routine clinical follow-up,

**Table 2. Levels of prevention and care levers used in the simulations.**

| Prevention | | |
|---|---|---|
| Linkage to PrEP | Adherence to PrEP | Retention of PrEP |
| (0) 7% individuals once testing negative are linked to PrE | (0) Individuals are distributed as follows: 21% is non-adherent / 7% is low-adherent / 10% is high-adherent / 62% is full-adherent (mean risk reduction = 69.1%) | (0) Individual have a 53.5% chance of staying on PrEP each year |
| (1) 2% annual increase in the chance of linking to PrEP once testing negative | (1) 0% / 14% / 17% / 69% (mean risk reduction = 84.3%) | (1) Annual chance of retention for PrEP is increased to 65.0% |
| (2) 4% annual increase in the chance of linking to PrEP once testing negative | (2) 0% / 0% / 0% / 100% (mean risk reduction = 95.0%) | (2) Annual chance of retention for PrEP is increased to 75.0% |
| (3) 6% annual increase in the chance of linking to PrEP once testing negative | | (3) Annual chance of retention for PrEP is increased to 85.0% |
| (4) 8% annual increase in the chance of linking to PrEP once testing negative | | |
| (5) 10% annual increase in the chance of linking to PrEP once testing negative | | |
| Treatment | | |
| **Linkage to ART** | **Viral suppression** | **Retention in ART care** |
| (0) 86% of those tested are linked | (0) The chances of becoming suppressed are; 50.3% for the first visit, 22.1% for the second visit, 10.6% for following visits (83.3% chance of suppression after 3 visits) | (0) 90% of individuals are retained across care visits (72.4% for 3 care visits) |
| (1) 100% of those tested are linked | (1) These rates are increased by 2% annually (After 9 years, the chance of suppression after 3 visits | (1) 2% increase in the retention rate across visits (77.4% of individuals are retained for 3 care visits) will be 95%) |
| | (2) These rates are increased by 3% annually (After 6 years, the chance of suppression after 3 visits will be 95%) | (2) 5% increase in the retention rate across visits (85.2% of individuals are retained for 3 care visits) |
| | (3) These rates are increased by 5% annually (After 4 years, the chance of suppression after 3 visits | (3) 10% increase in the retention rate across visits (96.7% of individuals are retained for 3 care visits) will be 95%) |

generally 3 months [50], requiring retention in regular care to be maintained. If a lapse in retention occurs, the individual will no longer have access to PrEP medication. We use RADAR cohort data capturing whether a participant receiving PrEP at the time of interview as a proxy to estimate the baseline level of PrEP retention (53.5% retention annually) and use the PrEP retention lever to perturb this rate for 3 additional levels. At level 1 annual retention is 65%; at level 2 it is 75%; and at level 3 it is 85%.

**Linkage to HIV care.** Linkage captures the extent to which people newly diagnosed with HIV or who have fallen out of care are linked to HIV care to initiate ART. Linkage to care is measured as having a CD4 or viral load test after diagnosis. CDPH data show that levels of linkage at baseline are high (86%). Initial experiments showed limited sensitivity of the model to small increments in increasing linkage; therefore we retained only a single potential perturbation in which linkage is increased to 100%.

**Viral suppression.** The effectiveness of the TasP strategy depends on achieving and maintaining an undetectable viral load. Most people will achieve an undetectable viral load within 5 months of starting ART [51]. There are, however, cases in which suppression is not achieved due to intra-host biology preventing the drug from taking effect or failure to adhere to medication. Based on CDPH data LHM uses the percent virally suppressed over continuous care visits at or above the baseline level. The 3 levels for viral suppression beyond baseline consist of 2%, 3%, or 5% annual increases.

**Retention in HIV care.** Retention in HIV care is key to maintaining viral suppression [52]. CDPH surveillance data suggest that regular meetings occur roughly every 3 months, and that the chances of being retained across these visits is on average 90% for the baseline level (this varies by race/ethnicity, age, and length in care; see S1 Appendix). Using the retention in HIV care lever, we increase this rate of retention from 1 visit to the next by 2%, 5%, or 10% for levels 1, 2, and 3, respectively (capping at 100%).

**HIV testing.** Diagnosis, or frequency of testing, is commonly identified as a lever in the HIV prevention system. We exclude it from our experiment because testing rates are proprietary and unavailable to CDPH for monitoring. We have, however, conducted supplemental simulations to examine the sensitivity of our findings to changes in this rate in our sensitivity analysis section.

**Size of the simulation experiment.** This experimental setup resulted in 2304 ($6*3*4*2*4*4$) combinations of levers, or scenarios. To account for variation in the model outcomes and gain more reliable results, we simulated each scenario 44 times. This number of replications provides sufficient reliability to construct 95% CIs for estimating a scenario's population mean incidence rate. This number of replications also allows us to display meaningful variations in these rates within single or combined scenarios. In our figures and tables, we present 80 percentile simulation intervals as these quantiles are relatively stable even if based on a single scenario with 44 replications. For each run we use an initial burn-in period of 5 years and measure the annual incidence over time for 15 years after burn-in, representing the period from 2016 to 2030.

## Analysis of experimental data

After announcing the GTZ plan, Chicago launched its EHE initiative in 2018, which aims for 90% reduction of new HIV infections by 2030. Unlike the national plan, the local plan does not mention an intermittent goal of 75% reduction of new infections by 2025, despite it being of interest to local policy makers. In 2018 there were 564 new HIV infections diagnosed among MSM in Chicago; to attain EHE goals, Chicago would have 141 new cases in 2025 and 56 cases in 2030. We assume that any simulation with fewer observed cases at 2025 or 2030 respectively will have "gotten to zero".

Because we anticipate that the most successful scenarios that meet these criteria will involve combinations of higher levels of several levers of change, we use partition, or classification, trees [53] to analyze which combinations of levels in each lever achieve 75% or 90% HIV infections reduction by 2025 or 2030, respectively. Our classification tree analysis uses this binary criterion as the dependent variable for each of the 101,376 runs—that is 44 replications of all

2304 scenarios—with predictors being the levels of each lever. Classification trees form a tree structure based on an algorithm that searches for an optimal split of a node among all possible high and low dichotomizations of each predictor, e.g., 0 or 1 versus 2+ for retentions in ART care. Such a split forms 2 new nodes. A tree is "grown" by subjecting each new terminal node to this same search for the lever and cut point that optimally divides each new node into 2 additional nodes, representing those that are closest to meeting the reduction criterion and those that do not. All possible splits are evaluated at each node, and the split is determined based on minimizing the Gini index:

$$gini = 1 - \sum_{i=1}^{n}(p_i)^2 \qquad (1)$$

where $p_i$ is the proportion of each node attaining the EHE goal. This algorithm continues to grow the tree by using the splitting algorithm at each terminal node, thus partitioning the entire space of scenario runs into locally determined splits. After growth, the tree is "pruned" of unstable end nodes using cross-validation. By following tree "branches" of better performing splits, the higher-performing levers and levels making the best progress can be displayed. We use the binary outcomes of success for each run to create a partitioning tree in which the most efficient pathways to achieving the EHE goals for Chicago are represented. Computations and figures were performed in R using the functions rpart and rpart.plot [54].

## Using a bayes predictive distribution to project from the most recent incidence rate history

To examine whether Chicago's most recent surveillance data is close to reaching the EHE goals, we used the Bayesian posterior predictive distribution [55] of the reduction in HIV infections by 2030 based on yearly MSM incidence from 2016 through 2019, the latest year with epidemiological data available from CDPH. This projection weights all trajectories based on how well they match the 2016–2019 rates and assesses what progress can be expected in the next 15 years should current patterns continue. We use pseudo-classes [56–58] to obtain a non-parametric posterior predictive distribution of new HIV infections among MSM in 2030. This pseudo-class method provides asymptotically unbiased estimates of this predictive distribution (see S2 Appendix for details). We report the mean and median of the predictive distribution and the likelihood of attaining the EHE goals.

In Addition, we compared attaining EHE goals with a status neutral approach, which supports both HIV positives and negatives simultaneously, to 2 alternative approaches; one that increases PrEP linkage, adherence, and retention while keeping the three ART components at their baseline levels, and one that increases ART linkage, viral suppression, and retention while keeping PrEP components at their baseline levels.

## Results

To check the systemic behavior of the model, we present the model validation using 1-year (2016) projections of new HIV infection overall, by age, and by race/ethnicity group.

### Model validation

**Total Annual HIV Incidence** Per data collected by the CDPH, a total of 665 new HIV cases occurred among MSM in Chicago in 2016. Over the 44 model runs with baseline behavior (without additional interventions), LHM predicted a mean of 690.9 new HIV cases for 2016

(95% CI, 647.8—733.7). The observed incidence is within the confidence intervals of our mean model behavior, showing that LHM accurately reflects the observed HIV infection rates.

**Annual HIV Incidence, Distributed by Age** Fig 2 shows the modeled distribution among each of the CDPH's chosen age groups (grey) and how it compares to data from 2016 (blue). Table 3 compares the mean and 2 simulation-based intervals with the observed HIV infection rates by age group for 2016. The 80% simulation prediction interval corresponds to a range of 1-year age-specific HIV infections based on the 44 simulations of the baseline levels (i.e., the $10^{th}$ and $90^{th}$ prediction percentiles over the 44 replicates). We also provide 95% CIs for mean population-based, age-specific incidence rates. Unlike the former intervals, the size of these latter intervals decreases with the number of replications and gives accurate bounds for the mean trajectory.

These results show that the model presents an age distribution that closely approximates the age distribution of CDPH surveillance data for 2016. It shows a peak of new HIV cases in the 20–29 age bracket, and a decline towards the older age ranges. Because we observe only a slight overestimation of incidence, primarily in the 13–19 age bracket, we consider the model's distribution of incidence by age a reliable representation of the real world.

**Annual HIV Incidence, Distributed by Race/Ethnicity** Consistent with CDPH reporting we examine incidence among MSM for 4 race/ethnicity groups (Non-Hispanic Black, Non-Hispanic White, Hispanic, Other). For each of these groups we compared the observed rate of incidence based on 2016 CDPH data, with rates produced by our model. The result (Table 4) show that the model must consider community-level factors, including community viral load and local hardship, to reproduce observed disparities. Therefore we included the health disparities module, which resulted in prediction intervals that propperly reflect rates observed in the real world. Even with this module in place there was a 31% overestimation of the mean incidence among MSM in the 'Other' race/ethnicity group. However, only 5% of the population falls into this "Other" category, and therefore consider this difference to have a very small effect on projections. Additionally, there is a modest 16% underestimation for rates for the non-Hispanic Black MSM, and overestimation of non-Hispanic white and Hispanic MSM by 8% and 24% respectively. Overall, these results indicate that the LHM results fit the observed incidence by race/ethnicity moderately well, but there remains room for improvement. Because this article discusses only the population level impacts, and because the LHM (with the health disparities module) produces realistic racial incidence distributions, we deem it adequate for our experiment.

## Experimental results

Herein we examine the complete 15-year period included in the EHE plans using the previously described experimental design. The resulting partitioning trees show the vast majority of scenarios do not attain the year 10 (2025) (Fig 3) and year 15 (2030) EHE goals (Fig 4). For 2025 (Fig 3), the 21% in the top primary node, indicates the percentage of all scenarios that reach 75% reduction goal by 2025. Similarly, the 6% in the top primary node in in Fig 4 indicates the percentage of all scenarios that reach the 90% reduction goal by 2030. The best splits are identified down each path, with greater impact on incidence reduction on the left and lesser impact on the right. For example, 35% of scenarios with PrEP linkage to care ≥3 achieve 75% reduction by 2025 years whereas 7% of scenarios with PrEP linkage to care <3 achieve this goal. Dark red nodes represent very low rates; lighter red, low rates; lighter blue, modestly high rates; and darker blues, high rates.

Three distinct pathways yielded a reasonable likelihood of success at at reading 75% reduction by 2025 (Fig 3). The path with the greatest proportion of runs (77%) achieving the goal

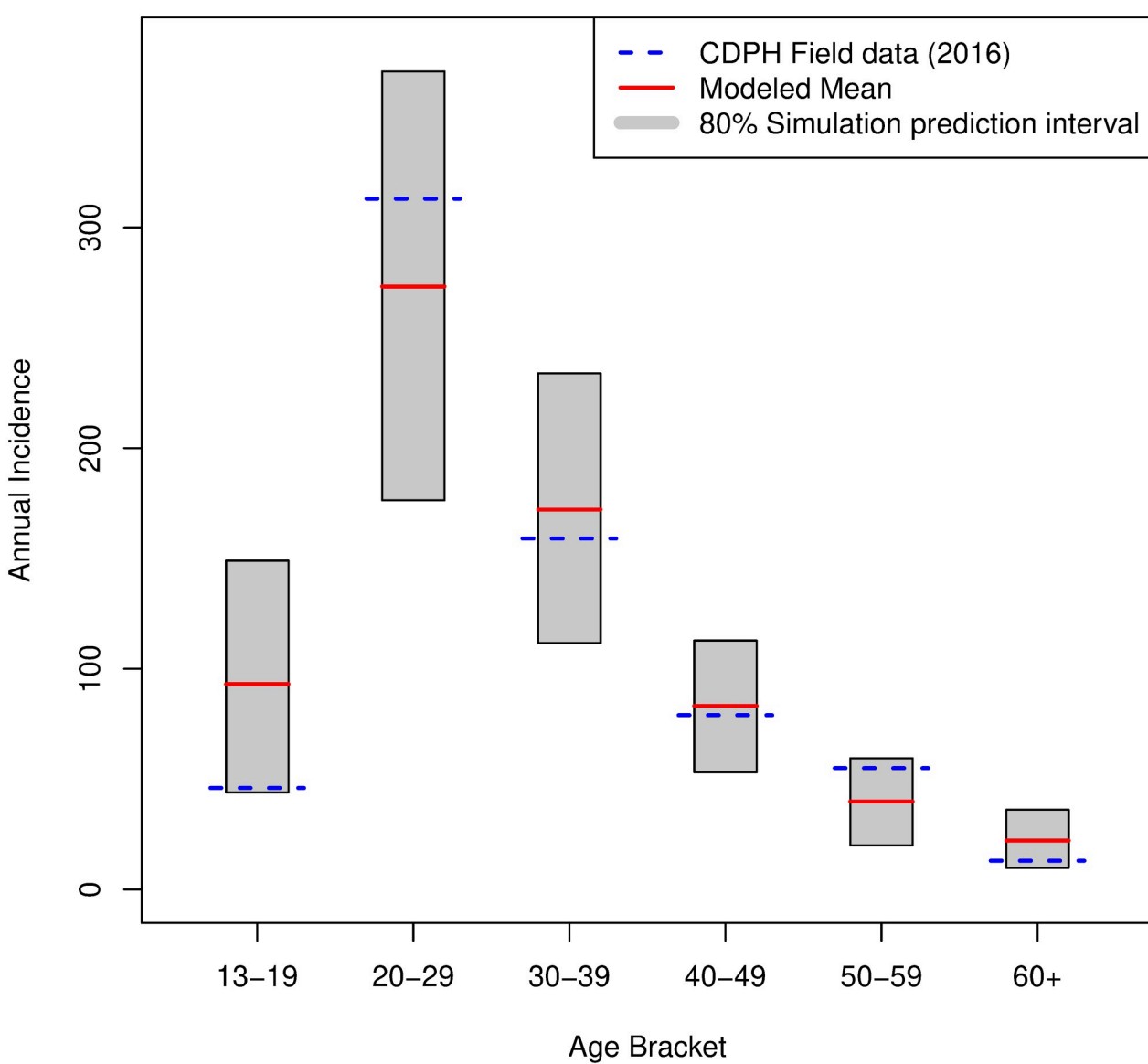

**Fig 2. Incidence by age: Observed and predicted annual incidence cases by age, the modeled 80% prediction range (grey), the modeled mean value (red), and the count observed in 2016 field data from the Chicago Department of Public Health.**

(Path 1, bold black line) entails increases in PrEP linkage (level≥3), PrEP retention (level≥2), and PrEP adherence (level 2). The second most efficient path (68%)(Path 2, dotted line) entails increases in PrEP linkage (level≥3) and PrEP retention (level≥2), and allows PrEP adherence levels of <2. Path 2 includes the baseline unperturbed level and it does not require a specific level of PrEP adherence. Instead, it requires the highest level of ART retention, and reaches its goal only by combining treatment and prevention levers. The third most efficient path (63%) (Path 3, dashed line) entails increases in PrEP linkage (level≥3), PrEP adherence (level 2) and ART retention (level 3). Unlike Path 1 and 2, Path 3 does not require any specific levels of PrEP retention. Compared to the best path, Path 1, Path 3 permits lower levels of PrEP retention but requires maximum ART retention, yielding a 14% reduction (63% vs. 77%) in the proportion of runs attaining the 2025 goal.

**Table 3. Observed and predicted annual incidence cases by age.**

| Age Group | 2016 Field data (CDPH) | Modeled prediction: Mean | Modeled Prediction: 80% simulation prediction interval | Modeled Prediction: 95% CI of the mean |
|---|---|---|---|---|
| 13–19 | 46 | 93.1 | 44.0–149.0 | 80.6–105.5 |
| 20–29 | 313 | 273.2 | 176.4–370.8 | 249.2–297.3 |
| 30–39 | 159 | 172.2 | 111.7–234.0 | 157.6–186.7 |
| 40–49 | 79 | 83.2 | 53.1–112.8 | 75.5–90.9 |
| 50–59 | 55 | 39.9 | 20.0–59.4 | 35.1–44.6 |
| 60 + | 13 | 23.1 | 9.8–36.1 | 18.5–25.8 |
| Total | 665 | 690.8 | 519.7–827.7 | 647.8–733.7 |

**Notes:** CDPH indicates Chicago Department of Public Health.

For 90% reduction by 2030 (Fig 4) only one path has an acceptable chance (58%) of reaching the goal. This path consists of the highest level of ART retention, moderate levels of PrEP linkage (level≥2), high levels of PrEP retention (level≥2), and the highest levels in of PrEP adherence (level 2). Compared with Path 1 for 2025, this path adds a requirement of maximum ART retention, while reducing the required PrEP linkage (from level≥3 to level≥2); however, a much lower proportion of runs achieves the desired 2030 reduction goal (58% vs. 77%).

This optimal path can be translated into an HIV prevention and care cascade to be used for EHE prevention and care goals based on different combinations of prevention and care levels of linkage, retention and adherence. Fig 5 displays projected 2030 cascades and compares cascade steps assuming continuation of levels from 2015 vs. the optimal path for the 2030 goal. For the 2030 goal, the combination of PrEP and ART levels identified in the optimal path resulted in a 41% higher rate of viral suppression among MSM as compared to continuation of baseline levels (75.7% vs. 53.75%). Similarly, we observe increases (490%) in the percentage of eligible MSM receiving PrEP and a consequent reduction of individuals at risk.

## How well are current trends progressing towards EHE goals?

Fig 6 shows incidence projections for the optimal path (bottom curve, green) and baseline path (top curve, blue) and compares them with the most recent CDPH surveillance data for new HIV diagnoses (2017—2019)(black curve). The CDPH data shows that the number of

**Table 4. Observed and predicted rates of HIV incidence for MSM by Race/Ethnicity.**

| | NH Black | NH White | Hispanic | Other |
|---|---|---|---|---|
| **Observed values (based on CDPH data)** | | | | |
| Incidence rate per 10k | 183.3 | 51.8 | 91.8 | 47.8 |
| **LHM without the Health disparity Module** | | | | |
| 80% Prediction Interval | 57.1–122.1 | 66.0–117.1 | 70.1–142.8 | 21.2–122.3 |
| Mean incidence rate | 88.5 | 89.7 | 109.1 | 68.2 |
| Relative error | 51.7% | 73.4% | 18.9% | 42.9% |
| **LHM with the Health disparity Module** | | | | |
| 80% Prediction Interval | 110.5–202.5 | 42.9–74.0 | 76.0–148.0 | 20.7–103.5 |
| Mean incidence rate | 153.4 | 56.0 | 114.1 | 62.6 |
| Relative error | -16.3% | 8.3% | 24.3% | 31.0% |

**Notes:** NH indicates non-Hispanic.

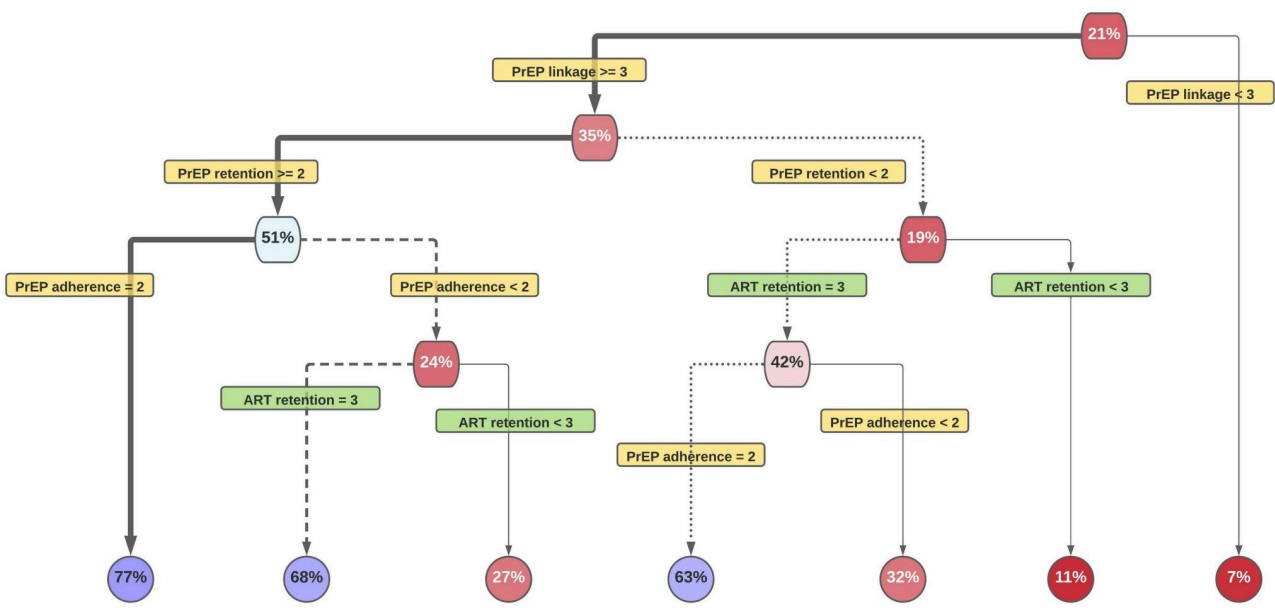

**Fig 3. Pathways toward interim EHE goal, 75% reduction of incidence by 2025.**

HIV infections has steadily declined for the overall MSM population. While still within the 80% prediction range of our baseline model, the trend suggests gains beyond what would be expected from continuation of 2015 baseline care and prevention activities. Using the observed trajectory of yearly infections from 2016 through 2019 we calculated a Bayesian posterior predictive distribution [55] of the reduction in HIV incidence by 2030 and assessed if the current trend would meet the EHE goals. The results of this projection (Fig 7)(black curve) show that despite improvements every year, the continuation of current trends would not meet EHE

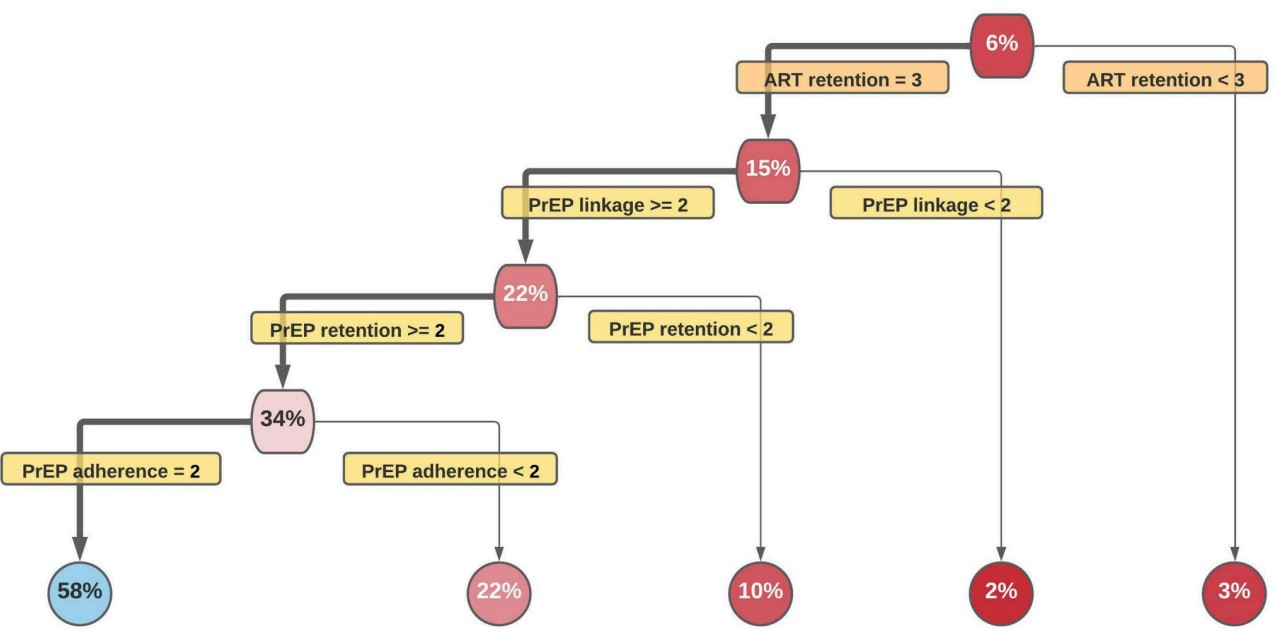

**Fig 4. Pathways toward final EHE goal, 90% reduction of incidence by 2030.**

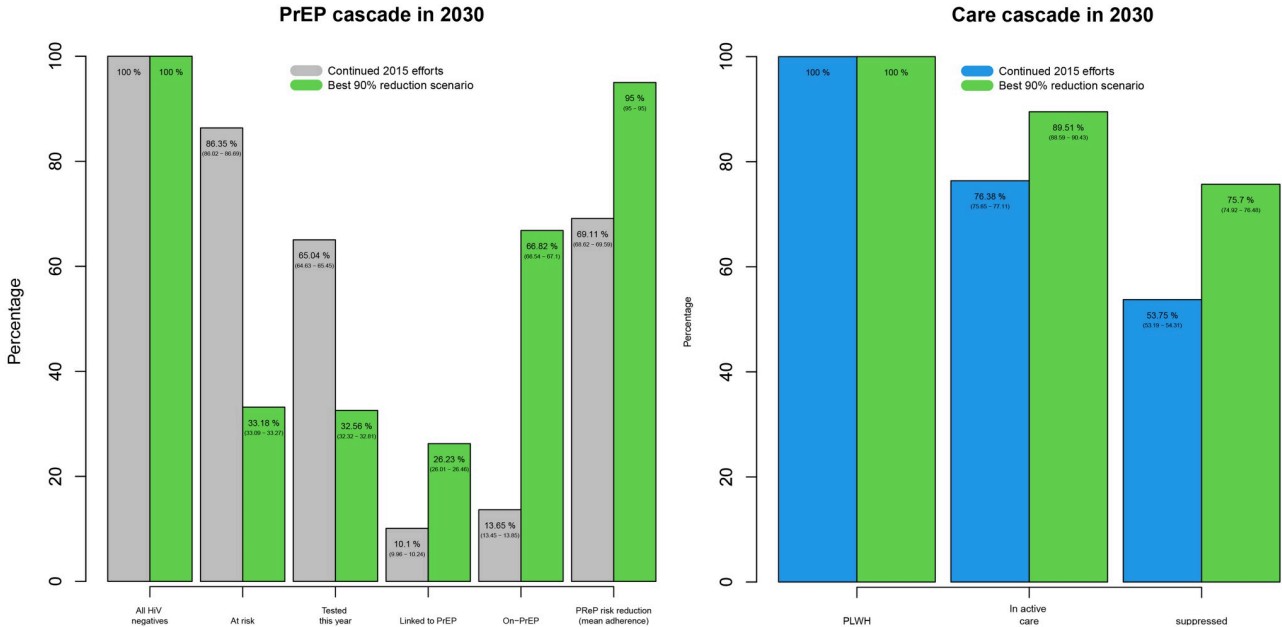

**Fig 5. A comparison of care cascades between continuation of 2015 baseline levels vs. optimal path toward the 2030 goal of 90% reduction, for both prevention (left) and treatment (right).**

goals. Extrapolation of the current trajectory yields 358 cases by 2025 (a 37% reduction), and 313 in 2030 (a 45% reduction). Based on the quantiles of this distribution the current trajectory has a 0.013 likelihood of reaching 75% reduction by 2025 and a 0.0006 likelihood of reaching 90% reduction by 2030.

Fig 8 shows the projection of the optimal pathway to the 90% reduction goal and 2 pathways that use either the maximum prevention (PrEP) or care (ART) levers in isolation. The PrEP-alone improvement scenario is a pathway in which only the PrEP levers are maximally increased, and ART-alone improvement scenario similarly maximizes the levers for ART while leaving PrEP levers untouched. Neither of these scenarios achieved the levels of reduction attained by the optimal combination strategy, but the PrEP-alone scenario gets close.

## Robustness of results

To ensure the robustness of our results, we explored the sensitivity of our main model outcomes (the decision tree for achieving 90% reduction by 2030) for variations in the 2 inputs for which we have relative uncertainty: "treatment discontinuation rate" for those receiving care for longer than 2 years, and the "rate of testing". First, we compared our main decision tree with two alternative fitting values parameters of "treatment discontinuation rate". This comparison (S3 Appendix) revealed minor differences in the order of splits and the overall rates of success depending on the fitting method used; however, the pathways to attaining the desired reductions of annual incidence remain nearly identical. As such, we conclude that our model provides a robust set of core lever combinations required to achieve the reduction goals. Next, we compared modeled outcomes for different rates of testing. In the absence of detailed local data on testing rates, our model assumed behaviors to follow uniform rates of 77.1% of individuals receiving annual testing [59], a rate that we vary in the second set of robustness checks. The results of this analysis (S4 Appendix) show that increases in the rate of

## Modeled incidence over time

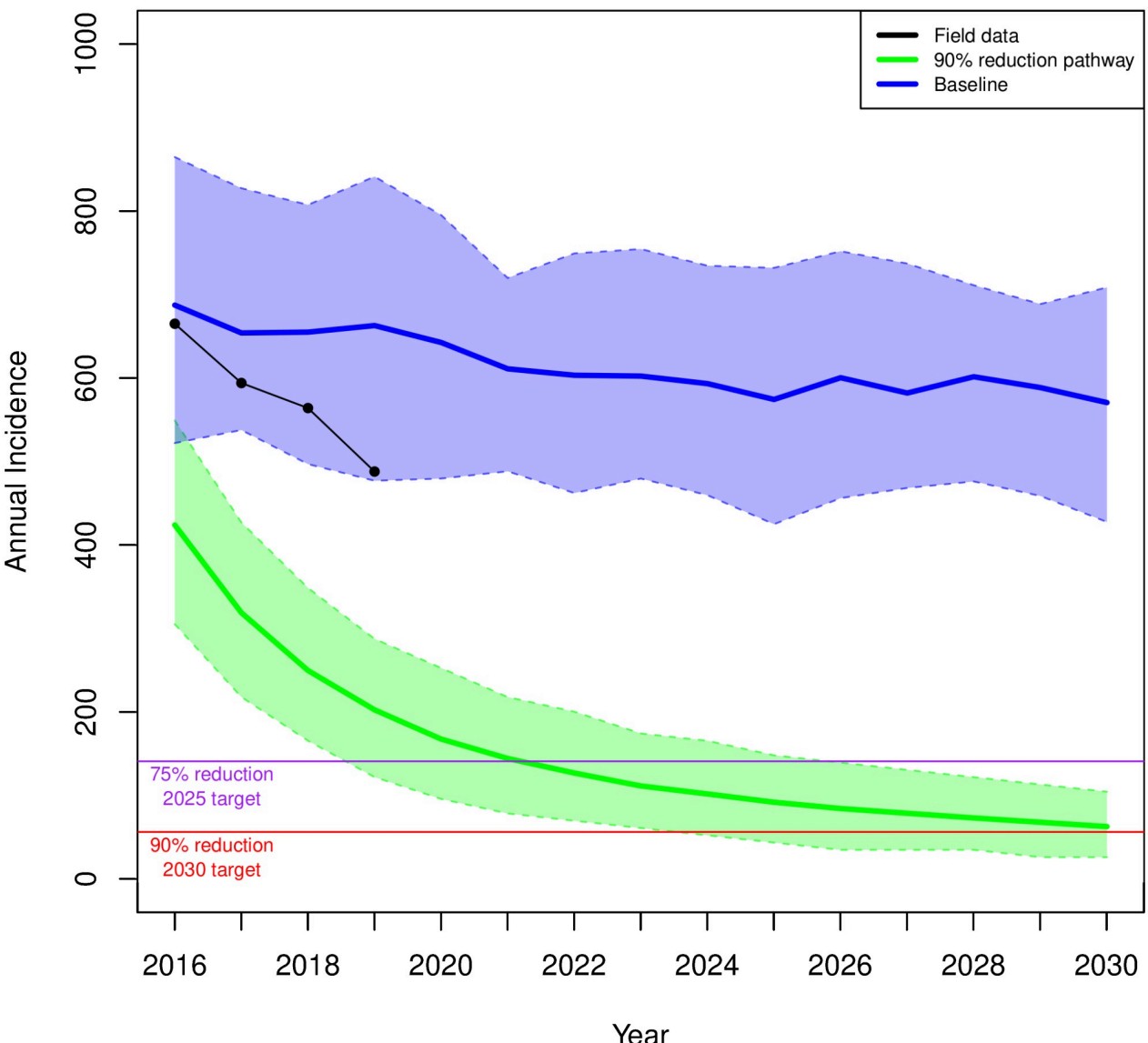

**Fig 6. Projected incidence over time: Projected number of incidence cases over time for the baseline model (blue) and the optimal 90% reduction scenario (green).** Solid line shows the mean projection; colored surface, the 80% Prediction Interval. The black line shows observed levels based on the most recent CDPH surveillance data.

testing yields minor differences in the order of splits and the overall rates of success. However, the pathways to attaining the desired reductions of annual incidence again remain stable. An exception is that extreme increases in the rate of testing can negate the need for increases in retention in HIV care. This is not a surprising observation, because in our model, testing for HIV-positive individuals is synonymous to their potential to re-engage in treatment, and those who are already HIV positive will re-engage in treatment after receiving a positive test result.

## Modeled incidence over time

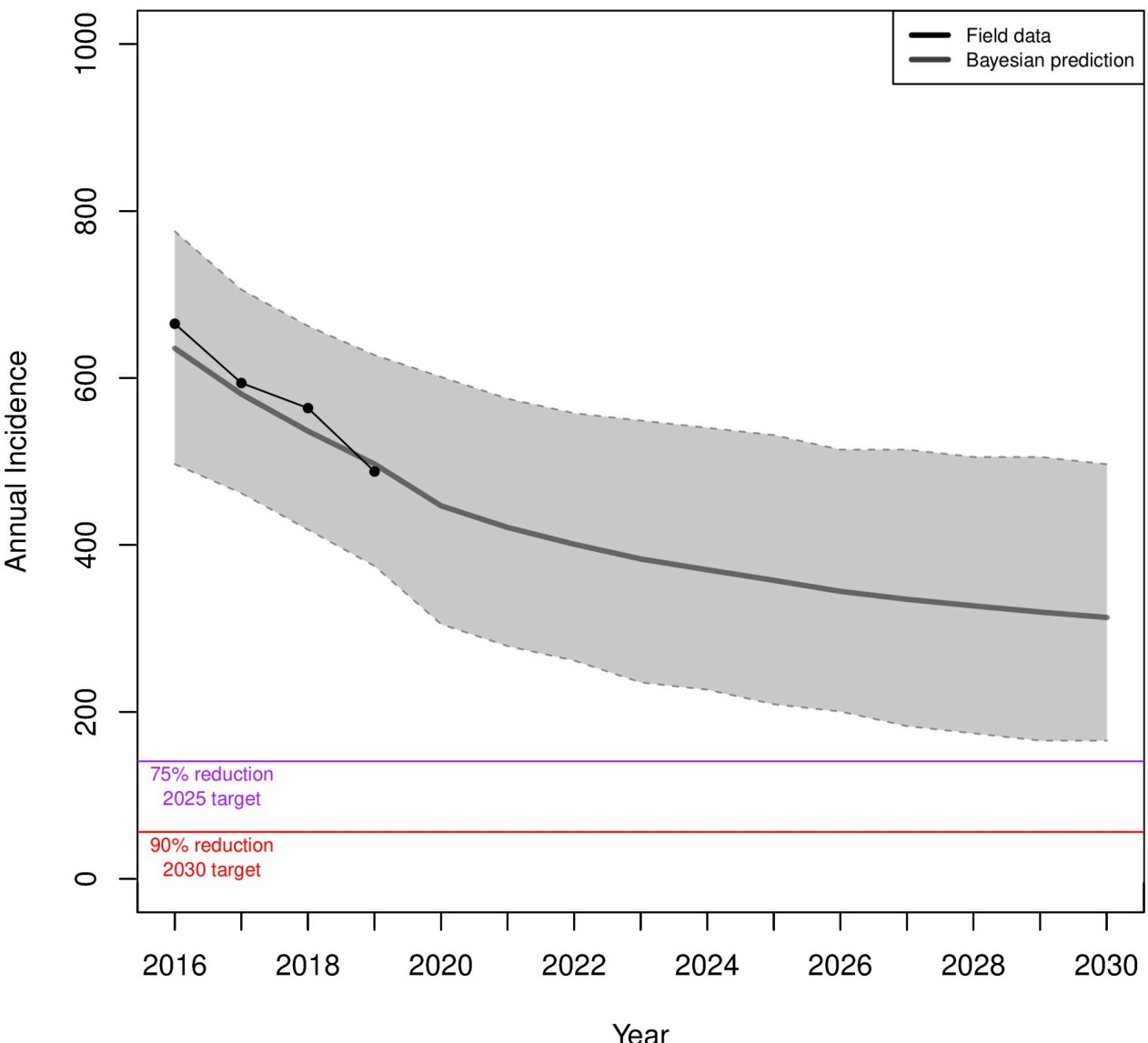

**Fig 7. Projected number of incidence cases over time based on Bayesian posterior predictive distribution: Solid line shows the mean projection; colored surface, the 80% Prediction Interval.** The black line shows observed levels based on the most recent CDPH surveillance data.

## Discussion

Our model results suggest that the GTZ goal of a 90% reduction in new HIV diagnoses by 2030 will require considerable scaling up of interventions and strategies in both the prevention and care cascades. Our results highlight the importance of a combination of retention in HIV care and the central role of PrEP, at full scale, to reach this goal. Our findings align with previous observations made by Shah and colleagues [7] and Jenness et al [8] in that all models suggest limited impact of retention in HIV care in isolation. Similarly, our findings support the claims made by Khanna et al [28] that neither linkage to PrEP car nor retention in PrEP care

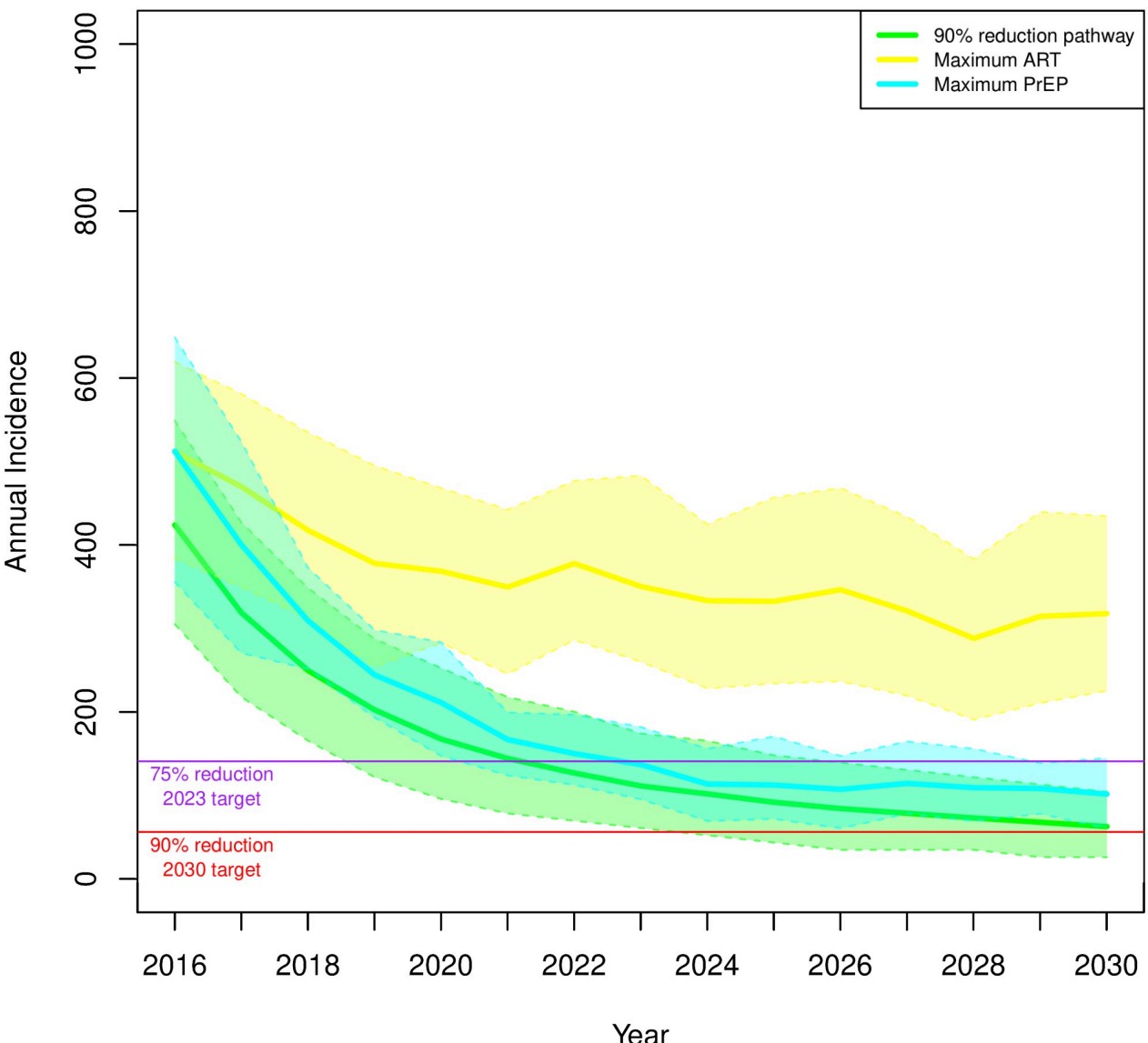

**Fig 8. Projected number of incidence cases over time for extreme scenarios: Projected incidence over time for 90% reduction (green); the ART-only scenario that maximizes only ART related levers (yellow); and the PrEP-only scenario that maximizes only PrEP related levers (purple).** The solid line represents the mean of each projection; colored surface, the 80% Prediction Interval.

in isolation is likely to achieve GTZ goals. Our experimental design that encompasses both treatment and prevention simultaneously provides a comprehensive overview of what combination of interventions is required to "get to zero". In contrast to previous work [7, 8], we find that complementing retention in treatment (ART) with increased testing has only limited impact, and that the path to substantial reduction in new HIV diagnoses and "getting to zero" necessitates substantial efforts in prevention using PrEP with increased levels of retention in treatment. Our model results suggest that pathways with a strong likelihood of attaining the GTZ goal at minimum entail an annual increase of 4 percentage points in the percent of people

linked to PrEP care (from 7% to 47% by 2025 and 67% by 2030); 75% or more of people receiving PrEP being retained in PrEP services annually; all people receiving PrEP to be fully adherent; and 96.7% of people with HIV to be retained for 3 care visits after diagnosis. While we find that the PrEP cascade has more room for improvement than the care cascade, and hence seems to have a larger potential to move the needle initially, only a combined effort that includes increases in retention in HIV care has a reasonable chance of success when it comes to achieving the GTZ goals. Similar to the study by Nosyk et al [9], our results support and stress the considerable contribution of a status neutral approach [60].

The main goal of GTZ is to reduce new incident cases by 90% by 2030, and to achieve this goal this plan highlights a multi-sector collaboration focusing on two sub-goals: (1) increase by 20 percentage points the number of people living with HIV who are virally suppressed, and (2) increase by 20 percentage points the number of people vulnerable to HIV who use PrEP. In considering the cascades of treatment and care produced by the scenario that is most efficient in attaining the main GTZ goal we found these sub-goals poorly aligned with the main GTZ goal. Our optimal scenario far exceeds the suggested 20 percentage points increase in people vulnerable to HIV who use PrEP. What is more, this scenario failed to reach the first sub-goal and did not yield the suggested 95% viral suppression levels. Our model results also suggest that an optimal intervention approach may reach the main GTZ goal without reaching 95% viral suppression. In doing so, our results assess the potential success of a set of interventions, can inform evaluation of ongoing progress, and identify critical sub-aims within the GTZ initiative, all of which are critically important for policy makers.

While mostly consistent with previous HIV modeling work in different contexts, our model's focus specifically on MSM in Chicago presents a challenge as to the generalizable of our findings. While the local context is likely to impact the numerical results of interventions, the robustness of the found optimal path to reducing the epidemic suggests that even without numerically identical outcomes our work contributes to the body of work (e.g., [9]) that highlights the need for a status neutral approach to end the HIV epidemic. Similarly, our focus specifically on unprotected sex among MSM prevents us from drawing a comprehensive picture of the system-level HIV dynamics in Chicago. However, the fact that this model accounts for roughly 70% of new incidence cases in Chicago makes us confident that our identified path extends well beyond this population, while we acknowledge that numerical values will likely differ in a full population approach. Similarly, while numerical results are likely to vary outside our context, the robustness of our pathways to "getting to zero" is likely to persist and may be generalized across similar contexts.

## Limitations and future work

There are several limitations in our study. In interpreting our models results it is important to recognize that our model considers measurable levers of change rather than implementation strategies. As such, the question of which specific interventions, implementation strategies, and resources are needed to achieve these changes remains unanswered. Translating model outcomes into decisions about implementation strategies requires a mediation model that accounts for the complex systems involved [61]. Specifically, the levels used in our experiment are dimensional aspects devoid of knowledge about the intensity required to achieve these levels. For example, obtaining nearly perfect retention in HIV care can be modeled easily, but may be difficult to attain in practice. Therefore, when interpreting model findings, one should consider that an intervention target in the experiment, or a given level of intensity in modification, does not translate directly into an actionable approach in practice. While intervention levels in each lever are chosen to represent the range of what is theoretically possible, we did

not consider the practical implementation efforts required to attain these levels. Additionally, there are substantial differences between the 6 levers. The number of levels differ by lever, and the baseline values for each lever are different. Furthermore, the efforts required to move from one level to the next can differ greatly, not only across levers but also within a lever itself. For instance, facilitating linkage to care might be easier than maintaining someone in care, and increasing the percent retained in care from 50% to 51% is likely easier than improving it from 99% to 100%. When interpreting the model's results, it is crucial to be aware of this abstraction and use the model as an input to decision making rather than a replacement of that process altogether.

Additionally, as any model by definition is an abstraction, its power depends on its underlying assumptions. Being transparent in the validation and fitting process reveals limitations, boundaries, potential areas for further model development, and critical assumptions made in the model and as such has been an integral part of our reporting. Below we summarize the specific limitations related to the Levers of HIV model itself.

First, we found that our ART treatment module might have dynamics that were potentially incomplete, which would make it a prime candidate for updating. Our current mechanism is underspecified in terms of capturing the rates of dropping out of care (for those in treatment >2 years) and the process of re-engagement in care. While we have addressed this by considering a fitted term (for which we found no major impacts in our sensitivity analysis), we are currently in the process of refining this module based on more detailed local data from the CDPH pertaining to the care cascade.

Second, we found that racial disparities in incidence are only partially captured by our model. While we incorporated community-level factors that represent underlying structural factors and social determinants driving such disparities, fully explaining disparities will require additional refinement to the health disparities module. Although our model provides evidence that individual risk behaviors do not explain the existing disparities, integrating accurate mechanisms that drive disparities into our model is far from trivial. To open the black box of what is driving disparities and understand what interventions are needed to address them, a more comprehensive module should incorporate more detailed geo-spatial dynamics and other drivers of disparities such as stigma. It is a future aim for the research team and CDPH to better understand these disparities and to create more equitable intervention policy.

Third, it is important to note that our experimental design does not cover all interim steps in the prevention and care cascades. While the levers in our experiment cover "treat" and "protect" dimensions of intervention [9] a third relating to "diagnosis and screening" by means of testing is not completely covered in our experimental design. While an HIV test is a first step in (re)connecting to the care system, accurate local rates of HIV testing are scarce, and only linkage to care behavior is observed by local jurisdictions. As such, we opted to remove screening as a lever and instead check for potential impact of the testing rate as part of our sensitivity analysis, finding it to have limited impact on our modeling results.

Last, our model does not address exogenous factors that would either enhance or facilitate the achievement of high levels of the levers considered. Long-acting injectables, for example, could well promote adherence to PrEP and ART [62], making the high levels required to achieve 90% reduction more likely. Other exogenous factors could delay GTZ, such as the COVID-19 pandemic, which has diverted efforts away from meeting HIV goals in multiple locations [63] and is quite likely delaying GTZ progress. Such potential shocks to system dynamics are not considered in our model.

While our model has limitations, its value for decision support is notable. As George Box put it, "All models are wrong, but some are useful". Our model is particularly useful as a tool that provides evidence and concretely describes the level of scale-up needed to reach goals

in a situation where uncertainty is great. Although the current model does not provide the answer to a complex problem such as how to achieve the GTZ goals, its predictions can inform potential systemic change, and such information can be used as inputs for the discussion on what to target in intervention and implementation strategies. Further extensions of this model can inform which implementation strategies are likely or unlikely to meet or approach the levers needed to achieve the GTZ goals.

## Conclusion

We present a new high-fidelity ABM for decreasing HIV infections among MSM, focused on 6 levers of the prevention and care cascades. To support decision making by the local health department this model is aligned to the Chicago context, uses local individual level and community data, and has system behaviors that are validated against local HIV surveillance data.

Using detailed simulation experiments we found that achieving the GTZ goal of 90% reduction in HIV incidence by 2030 will require considerable and rapid scale-up of intervention efforts that go beyond those currently in place—beyond what has been envisioned as required to achieve this goal. Our findings suggest that achieving the GTZ goal for Chicago requires a status neutral approach that combines interventions in both prevention for HIV-negative individuals (through increased linkage, adherence, and retention to PrEP) and treatment for those who are HIV-positive (through increased retention in care). This approach can help policy makers plan and scale strategies needed to yield the greatest results on the HIV epidemic and achieve local and national goals to end it.

## Supporting information

**S1 Appendix. Complete Levers of HIV Model (LHM) description.** This appendix describes all details related to the implemented the LHM model based on requirement for model description set forth in the ODD protocol.
(PDF)

**S2 Appendix. Bayesian predictions.** This appendix describes in detail the Bayesian methods used for prediction in this paper.
(PDF)

**S3 Appendix. Robustness check of modeling outcomes—retention in care.** This appendix describes the details of the robustness check performed related to retention in care, and shows the experimental outcomes for two alternative ways of fitting the retention in care rates.
(PDF)

**S4 Appendix. Robustness check of modeling outcomes—rate of testing.** This appendix describes the details of the robustness check performed related to the rate of testing, and shows the experimental outcomes for two alternative rates of going in for HIV testing.
(PDF)

## Author Contributions

**Conceptualization:** Wouter Vermeer, Can Gurkan, Nanette Benbow, David Kern, C. Hendricks Brown, Uri Wilensky.

**Data curation:** Wouter Vermeer, Can Gurkan, Nanette Benbow, Brian M. Mustanski, C. Hendricks Brown.

**Formal analysis:** Wouter Vermeer, C. Hendricks Brown.

**Funding acquisition:** C. Hendricks Brown.

**Investigation:** Wouter Vermeer, Can Gurkan.

**Methodology:** Wouter Vermeer, Can Gurkan, Arthur Hjorth, Uri Wilensky.

**Project administration:** Wouter Vermeer.

**Resources:** Brian M. Mustanski.

**Software:** Wouter Vermeer, Can Gurkan, Arthur Hjorth.

**Supervision:** Nanette Benbow, Brian M. Mustanski, David Kern, C. Hendricks Brown, Uri Wilensky.

**Validation:** Wouter Vermeer, David Kern.

**Visualization:** Wouter Vermeer.

**Writing – original draft:** Wouter Vermeer, Can Gurkan, Nanette Benbow, C. Hendricks Brown, Uri Wilensky.

**Writing – review & editing:** Wouter Vermeer, Can Gurkan, Nanette Benbow, Brian M. Mustanski, David Kern, C. Hendricks Brown, Uri Wilensky.

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
