## [Decision Letter · Decision Letter 0]

14 Apr 2022

PONE-D-22-07578Prevention and Care Pathways to end the HIV Epidemic in Chicago: Projections of an Agent-Based Model for HIV among MSMPLOS ONE

Dear Dr.  Wouter Mermeer 

Thank you for submitting your manuscript to PLOS ONE. After careful consideration, we feel that it has merit but does not fully meet PLOS ONE’s publication criteria as it currently stands. Therefore, we invite you to submit a revised version of the manuscript that addresses the points raised during the review process.

We look forward to receiving your revised manuscript.

Kind regards,

Hamid Sharifi

Academic Editor

PLOS ONE

Journal Requirements:

Additional Editor Comments:

Dear Authors,

I read the manuscript and it could be for furthur consideration for publication in PLOS ONE. The main problem of the work is it has been written in a way that you cannot follow the text. I recommend to reshape the structure of the file and try to write it in a way to be readable for a wider researchers. You can move some parts of the text to Appendix.

Reviewers' comments:

Reviewer's Responses to Questions

**Comments to the Author**

1. Is the manuscript technically sound, and do the data support the conclusions?

Reviewer #1: Yes

Reviewer #2: Partly

2. Has the statistical analysis been performed appropriately and rigorously? 

Reviewer #1: Yes

Reviewer #2: Yes

3. Have the authors made all data underlying the findings in their manuscript fully available?

Reviewer #1: Yes

Reviewer #2: Yes

4. Is the manuscript presented in an intelligible fashion and written in standard English?

Reviewer #1: Yes

Reviewer #2: Yes

5. Review Comments to the Author

Reviewer #1: The paper highlights a critical issue profoundly and comprehensively. However, I have the following recommendations:

1. it is too long, they may shorten the introduction and even the other parts by dropping or summarizing less relevant sentences

2. the title of graphs and tables and the headers are not fully explained. For example, what is the scale of incidence in table 3 and figure 2?

3. the discussion is too short, and most of the main parts of the discussion were relocated in the conclusion. They have to organize their discussion accordingly. For example, limitations and the comparison between their results with others' findings are the main parts of the discussion

4. limitations and their impact on the final results are significant in such a deep and extended modeling. They have to explain this part in more detail and even paste a table to organize the main limitations and their possible impact on the final results. For instance, they only explored the MSM subpopulation, while it has a strong link with the other most at-risk sub-populations, which may change the results of this study if the protection and the prevention measures change simultaneously.

5. If they have access to the field data in more recent years to compare their projections with exact numbers to show the accuracy of their predictions.

Reviewer #2: This paper simulate the required changes in the local PrEP and ART cascade for meeting Chicago’s ’Getting to Zero’ goal of reducing new HIV diagnoses among MSM by 90% by 2030 using an agent-based model. They found that based on the current rate of decline (from 2016-2019) it is highly unlikely to achieve the 90% reduction target for 2030. In general, it can be said that valuable work has been done and deserves publication, however, the paper needs to be edited. The paper should be revised and edited by a professional editor to help improve the flow and its readability should be improved. I have some comments:

Introduction

1- In my opinion, the introduction section can be shorter and there are similar sentences in it, please summarize it.

2- Please be clear about the difference between your study and other studies that were conducted in Chicago and what the novelty of your study is.

Method

1- Please clarify that why you assume the proportion of MSM constant across age. Is it controllable in the ABM.

2- Please clarify that why you consider only 10 percent of LHM population.

3- Please clarify that why you not consider HIV testing as intervention in the model. Increasing testing frequency has the largest impact on reduction in HIV infections

Result

1- Please correct the symbols (e.g. <=) in the results text.

Discussion

1- It seems that the titles of the discussion and conclusion sections have been written in place of each other. Limitations should be in the discussion section.

2- In the discussion section, please compare your results with other studies that have used deterministic mathematical model and ABM.

Conclusion

1- The conclusion should be more concise. Please summarize it.

6. PLOS authors have the option to publish the peer review history of their article (what does this mean?). If published, this will include your full peer review and any attached files.

Reviewer #1: **Yes: **AliAkbar Haghdoost

Reviewer #2: No

---

## [Author Response · Author response to Decision Letter 0]

28 Jun 2022

Response to reviewer comments has been submitted as a separate file.

---

## [Decision Letter · Decision Letter 1]

25 Aug 2022

Agent-Based Model Projections for Reducing HIV Infection Among MSM: Prevention and Care Pathways to End the HIV Epidemic in Chicago, Illinois

PONE-D-22-07578R1

Dear Dr. Wouter Wermer

We’re pleased to inform you that your manuscript has been judged scientifically suitable for publication and will be formally accepted for publication once it meets all outstanding technical requirements.

Kind regards,

Hamid Sharifi

Academic Editor

PLOS ONE

Additional Editor Comments (optional):

Reviewers' comments:

Reviewer's Responses to Questions

**Comments to the Author**

1. If the authors have adequately addressed your comments raised in a previous round of review and you feel that this manuscript is now acceptable for publication, you may indicate that here to bypass the “Comments to the Author” section, enter your conflict of interest statement in the “Confidential to Editor” section, and submit your "Accept" recommendation.

Reviewer #2: All comments have been addressed

Reviewer #3: All comments have been addressed

2. Is the manuscript technically sound, and do the data support the conclusions?

Reviewer #2: Yes

Reviewer #3: Yes

3. Has the statistical analysis been performed appropriately and rigorously? 

Reviewer #2: Yes

Reviewer #3: Yes

4. Have the authors made all data underlying the findings in their manuscript fully available?

Reviewer #2: Yes

Reviewer #3: Yes

5. Is the manuscript presented in an intelligible fashion and written in standard English?

Reviewer #2: Yes

Reviewer #3: Yes

6. Review Comments to the Author

Reviewer #2: (No Response)

Reviewer #3: All issues raised by the Reviewers have adequately been responded to. This paper is important for care of PLHIV.

7. PLOS authors have the option to publish the peer review history of their article (what does this mean?). If published, this will include your full peer review and any attached files.

Reviewer #2: No

Reviewer #3: No

---

## [Editor Report · Acceptance letter]

6 Oct 2022

PONE-D-22-07578R1 

Agent-Based Model Projections for Reducing HIV Infection Among MSM: Prevention and Care Pathways to End the HIV Epidemic in Chicago, Illinois 

Dear Dr. Vermeer:

I'm pleased to inform you that your manuscript has been deemed suitable for publication in PLOS ONE. Congratulations! Your manuscript is now with our production department. 

Kind regards, 

on behalf of

Dr. Hamid Sharifi 

Academic Editor

PLOS ONE